# A low-nuclear Ag$_4$ nanocluster as a customized catalyst for the cyclization of propargylamine with CO$_2$

Lin Li[1,2,3,5], Ying Lv[1,2,3,5], Hongting Sheng[1,2,3] ✉, Yonglei Du[1,2,3], Haifeng Li[1,2,3], Yapei Yun[2,3], Ziyi Zhang[1,2,3], Haizhu Yu ®[1,2,3] ✉ & Manzhou Zhu ®[1,2,3,4] ✉

The preparation of 2-Oxazolidinones using CO$_2$ offers opportunities for green chemistry, but multi-site activation is difficult for most catalysts. Here, A low-nuclear Ag$_4$ catalytic system is successfully customized, which solves the simultaneous activation of acetylene (-C≡C) and amino (-NH-) and realizes the cyclization of propargylamine with CO$_2$ under mild conditions. As expected, the Turnover Number (TON) and Turnover Frequency (TOF) values of the Ag$_4$ nanocluster (NC) are higher than most of reported catalysts. The Ag$_4$* NC intermediates are isolated and confirmed their structures by Electrospray ionization (ESI) and [1]H Nuclear Magnetic Resonance ([1]H NMR). Additionally, the key role of multiple Ag atoms revealed the feasibility and importance of low-nuclear catalysts at the atomic level, confirming the reaction pathways that are inaccessible to the Ag single-atom catalyst and Ag$_2$ NC. Importantly, the nanocomposite achieves multiple recoveries and gram scale product acquisition. These results provide guidance for the design of more efficient and targeted catalytic materials.

The conversion of CO$_2$ into high-value-added chemicals[1-7], such as starch[8], carboxylic acid[9,10], propylene carbonate[11,12], and 2-oxazolidinone[13], is considered a promising approach to achieve carbon neutrality and has become a hot topic in the field of catalysis. In particular, 2-oxazolidone compounds have important application potential in organic intermediates, antibacterial drugs and chiral auxiliaries[14,15]. Ideally, the greenest preparation of 2-oxazolone compounds is the cyclization of propargylamine with CO$_2$. However, due to the unique structure of propargylamine, which contains both acetylene (-C≡C) and amino (-NH-) functional groups, it is difficult for most current catalysts to achieve this transformation[16-19]. Therefore, there is an urgent need to customize a catalyst with multiple active sites for the cyclization of propargylamine with CO$_2$.

Single-atom catalysts (SACs) have been widely used for CO$_2$ conversion due to their high molar utilization, clear active site, and unique electronic structure[20-23]. However, the presence of only a single metal site inherently limits SACs performance[24-28]. In contrast, low-nuclear nanoclusters (NCs) not only show the same characteristics as SACs but also benefit from synergistic effects between adjacent metals[29-36]. However, low-nuclear-weight NCs are more prone to unpredictable structural transformations under harsh environments[37,38], making it difficult to identify the true active component. Scott et al. reported that alkyne-protected Cu$_{20}$ NC do not require harsh pretreatment during catalysis[39], Wang et al. reported that an alkyne-protected Au$_{38}$ NC exhibited superior performance compared to that of a sulfate-protected Au$_{38}$ NC[40]. Zheng et al. found that the activity of intact Au$_{34}$Ag$_{28}$(PhC≡C)$_3$ is significantly better than that of partially or completely removed ligands[41]. Alkyne ligands, as metal-organic ligands, are considered to play an important role in improving the catalytic performance[42-44].

[1]Department of Chemistry and Centre for Atomic Engineering of Advanced Materials, Anhui University, Hefei 230601, China. [2]Key Laboratory of Structure and Functional Regulation of Hybrid Materials of Ministry of Education, Hefei 230601, China. [3]Key Laboratory of Functional Inorganic Material Chemistry of Anhui Province, Anhui University, Hefei 230601, China. [4]Anhui Tongyuan Environment Energy Saving Co., Ltd., Hefei 230041, China. [5]These authors contributed equally: Lin Li, Ying Lv. ✉e-mail: shenght@ahu.edu.cn; yuhaizhu@ahu.edu.cn; zmz@ahu.edu.cn

Therefore, we designed a low-nuclear Ag₄ NC protected by alkynes for the cyclization of propargylamine with $CO_2$. As expected, the customized Ag₄ NC achieved the highest TON value of 5746.2, significantly higher than that of reported catalysts and the corresponding Ag₂ NC, Ag₆ NC and Ag₉ NC. Moreover, three Ag₄ *NC intermediates were captured and confirmed their structures by ESI and ¹H NMR. The key role of four Ag atoms revealed the feasibility and importance of low-nuclear catalysts at the atomic level. More importantly, the obtained Ag₄/TNT nanocomposite afforded the product at the gram scale.

## Result and discussion

A low-nuclear alkyne-protected Ag₄ NC and the corresponding Ag₆ NC and Ag₉ NC were synthesized according to the literatures[45–47]. All these Ag NCs were characterized by mass spectrometry, UV–vis absorption spectroscopy, and single-crystal diffraction analysis (Fig. 1a and Supplementary Figs. S1–S3), confirming the atomic monodispersity and the exact formula assigned to Ag₄ NC, Ag₆ NC and Ag₉ NC, respectively. N-Benzylprop-2-yn-1-amine (1a, HC≡CCH₂NHBn) was selected as the preferred substrate for the cyclization of propargylamine to explore the catalytic performance of the customized Ag₄ NC. As expected, the Ag₄ NC protected by the acetylene ligand showed the best performance. To exclude the influence of the number of metal atoms, we designed and synthesized Ag₂ NC through a controlled experiment and compared their activity (Fig. 1a and Supplementary Fig. 4). Interestingly, among the Agₙ (n = 2,4,6,9) NC series, Ag₄ NC had the highest catalytic activity with TON and TOF values up to 5746.2 and 2873.1 h⁻¹, respectively, which were higher than those of reported catalysts (Fig. 1b and Supplementary Table 3). Then, we investigated the catalytic activity of $AgNO_3$, $AgBF_4$ and $[Ag(C≡C^tBu)]_n$, and the results show

that the activity of these catalysts is low. Furthermore, the Ag₄ NC with a Dppf (1,1′-Bis(diphenylphosphino)ferrocene) ligand was inactive for this reaction (Fig. 1b and Supplementary Table 1). The changes in the kinetics of the cyclization of N-benzylprop-2-yn-1-amine with $CO_2$ catalyzed by low-nuclear Ag₄ NC were monitored by in situ ¹H NMR (Supplementary Fig. 5a). Under the ideal conditions, we further explored the generality of the reaction for various propargylamine substrates. As shown in Fig. 1c and Supplementary Table 2, Ag₄ NC afforded the target products in high yields within 2 h for all propargylamine substrates (3a-4a) with alkyl terminations. Moreover, Ag₄ NC also reacted satisfactorily and afforded the corresponding products for substrates (5a-7a) with either electron-withdrawing or electron-donating groups. Most studies have reported that the low nucleophilicity of substrates such as N-phenylpropyl-2-yn-1-amine (2a) prevents the nucleophilic attack of carbon dioxide due to the benzene ring, resulting in a carbamate intermediate that is difficult to convert smoothly or requires high temperature conditions[16,48]. Much to our surprise and delight, the customized Ag₄ NC achieved highly active conversion of the N-phenylpropyl-2-yn-1-amine substrate at room temperature with yields up to 87%.

On this basis, we conducted relevant control experiments to gain more insight into the fundamental source of the catalytic activity of Ag₄ NC. The characteristic UV peak of Ag₄ NC showed a slight blueshift (8 nm) after Ag₄ NC was mixed with substrate 1a (1:2) for 1 h. In contrast, the characteristic peak of Ag₄ NC did not change at all within 2 h (Supplementary Fig. 5a). The adsorption of **1a** on the Ag₄ NC was detected by Fourier Transform Infrared (FT-IR) Spectroscopy. As shown in Supplementary Fig. 5c, the dominant stretching peak of C≡C-H at 3290 cm⁻¹ disappeared, and the peak of C≡C at 2106 cm⁻¹ shifted to 2120 cm⁻¹. This reveals that the H atom of C≡C-H was removed from

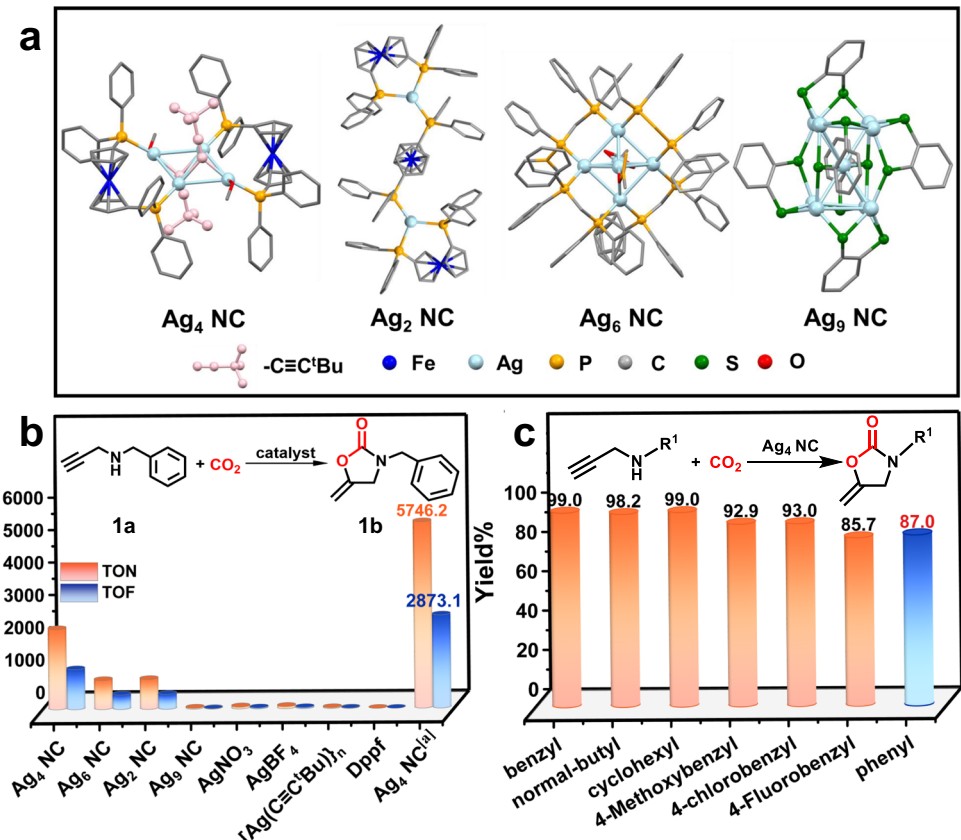

**Fig. 1 | Activity comparison and substrate expansion. a** Total structure of the Agₙ (n = 2,4,6,9) NCs. **b** TON and TOF value of different catalysts for $CO_2$ cycloaddition of N-benzylprop-2-yn-1-amine. Reaction conditions: Ag₄ NC (0.04 mol%), propargylamine (0.5 mmol), DBU (0.05 mmol), solvent (1 mL), 25 °C and $CO_2$ balloon.

Yields and selectivity were determined by gas chromatography. [a] propargylamine (1.5 mmol), DBU (0.15 mmol), solvent (1 mL), 25 °C and $CO_2$ balloon. **c** The cyclization of various propargylamine with $CO_2$.

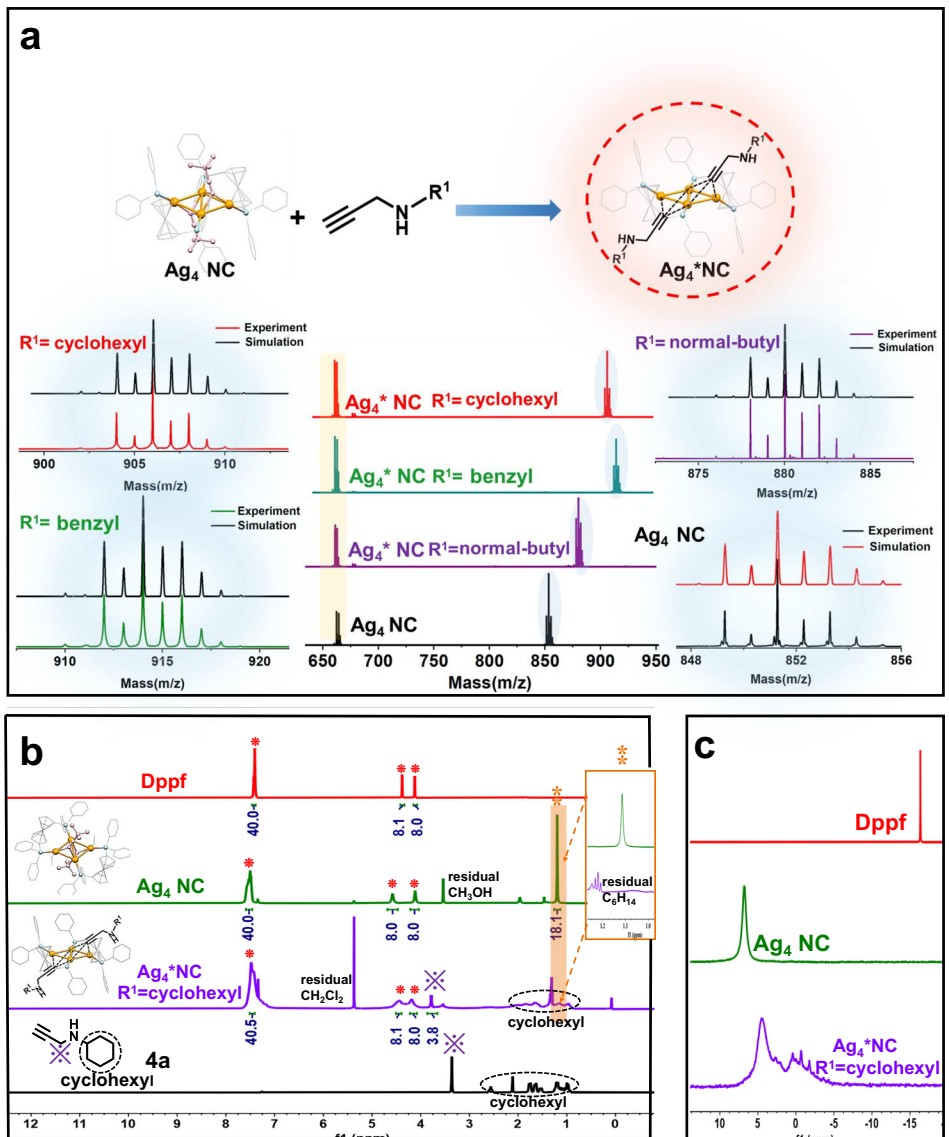

**Fig. 2 | Characterization of Ag$_4$ NC and Ag$_4$* NC. a** ESI-MS spectra of the intermediate Ag$_4$* NC and simulation of the corresponding mass spectrum. **b** $^1$H NMR spectra of **4a**, Ag$_4$* NC R$^1$ = cyclohexyl, Ag$_4$ NC, and Dppf. ✳ in red (Characteristic hydrogen of Dppf) ※ in purple (Characteristic hydrogen of the methylene group of **1a** and N-2-Propyn-1-ylcyclohexanamine) ※ in orange (The methyl hydrogen peak (1.11 ppm) of the C≡C$^t$Bu ligand disappears.) **c** $^{31}$P NMR spectra of Ag$_4$* NC R$^1$ = cyclohexyl, Ag$_4$ NC, and Dppf.

**1a** and that the C≡C bond of **1a** was activated by Ag$_4$ NC, which was related to the dehydrogenation activation of **1a**[49]. To obtain direct evidence of the interaction between Ag$_4$ NC and **1a**, we captured the Ag$_4$* NC intermediate by ESI-MS. As shown in Fig. 2a, the mass spectrum peaks of Ag$_4$* NC (R$^1$ = benzyl) were detected and calculated to be 661.0 m/z and 914.0 m/z (Simulation: 661.3 m/z = [Ag+Dppf]$^+$, 914.3 m/z = [1/2Ag$_4$-C≡C$^t$Bu + C≡CCH$_2$NHBn·CH$_3$OH]$^+$ respectively). The ESI-MS peaks of Ag$_4$* NC (R$^1$ = benzyl) corresponded exactly to those of Ag$_4$ NC (661.3 m/z and 851.3 m/z, Simulation: 661.3 m/z = [Ag+Dppf]$^+$, 851.3 m/z = [1/2Ag$_4$-CH$_3$OH]$^+$, respectively). Notably, Ag$_4$* (R$^1$ = benzyl) species were also successfully detected by ESI-MS in the reaction solution (Ag$_4$ + **1a** + CO$_2$), suggesting that Ag$_4$* NC is a key intermediate in the catalytic cycle (Supplementary Fig. 6a). To confirm this hypothesis, we isolated and verified the activity of Ag$_4$* NC (R$^1$ = benzyl). The experimental results showed that the activity of Ag$_4$* NC and Ag$_4$ NC was similar, confirming that Ag$_4$* NC was the key intermediate. The isolated Ag$_4$* NC (R$^1$=cyclohexyl), Ag$_4$ NC, dppf ligand and substrate **4a** were characterized by $^1$H NMR (Fig. 2b). The $^1$H NMR spectrum of Ag$_4$ NC contains the characteristic peak attributed to the hydrogen of the dppf

and tert-butylvinyl ligands, and the ratio of the intensities of the peaks attributed to the benzene ring hydrogen (7.49 ppm) on the dppf ligand to the metal ring hydrogen (4.49 ppm, 4.03 ppm) and the C≡C$^t$Bu ligand methyl hydrogen (1.11 ppm) was 40:8:8:18, and some peak shifts were observed. This was consistent with the molecular formula of Ag$_4$ NC, which reflects the structural integrity and high matching of the Ag$_4$ NC. Compared with the $^1$H NMR spectrum of **4a**, the $^1$H NMR spectrum of Ag$_4$* NC showed shifts in the characteristic peak of the hydrogen of the cyclohexyl group (marked by the black dashed circle) and the methylene hydrogen peak (purple symbol) in the substrate HC≡CCH$_2$NHCy (**4a**), while the methyl hydrogen peak of the C≡C$^t$Bu ligand (1.11 ppm) disappeared. Additionally, the ratio of the intensities of the peaks attributed to the methylene hydrogen of C≡CCH$_2$NHCy (3.69 ppm), the monocyclic hydrogen of the dppf ligand (4.10 ppm, 4.34 ppm) and the benzene ring hydrogen (7.40 ppm) was 3.8:8:8:40.5, indicating that the structure of the Ag$_4$* NC molecule was similar to that of the Ag$_4$ NC molecule, including two dppf ligands and two C≡CCH$_2$NHCy ligands. At the same time, it can be seen from the $^{31}$P spectrum (Fig. 2c) that the structure of Ag$_4$* NC is similar to that of

Ag$_4$ NC, and there is no free P ligand in the system. Moreover, the other substrates [HC≡CCH$_2$NHCy (**4a**, R$^1$ = cyclohexyl) and HC≡CCH$_2$NH$^n$Bu (**3a**, R$^1$ = normal-buty)] were selected for the primitive reaction with Ag$_4$ NC. The ESI-MS results showed two ionic peaks located at 661.0 m/z and 906.0 m/z (Simulation: 661.3 m/z = [Ag+Dppf]$^+$, 906.3 m/z = [1/2Ag$_4$-C≡C$^t$Bu+C ≡ CCH$_2$NHCy·CH$_3$OH]$^+$, respectively), along with peaks at 661.0 m/z and 880.0 m/z (Simulation: 661.3 m/z = [Ag+Dppf]$^+$, 880.3 m/z = [1/2Ag$_4$-C≡C$^t$Bu + C≡CCH$_2$NH$^n$Bu·CH$_3$OH]$^+$, respectively). Meanwhile, the Ag$_4$* (R$^1$ = cyclohexyl) species was also successfully identified in the reaction solution (Ag$_4$ + **4a** + CO$_2$) (Fig. 2a and Supplementary Fig. 6b).

Consistent with the experimental observations, the ligand exchange of 3,3-dimethyl-1-butyne (BH) with N-benzylprop-2-yn-1-amine (AH) was found to be thermodynamically feasible (exergonic by 6.4 kcal/mol, Fig. 3b). After that, two main types of mechanisms, depending on whether carboxylation occurs on the incoming A substrate (via ligand exchange, Path I) or an extra AH substrate (Path II, Supplementary Fig. 15 and Fig. 3b), were investigated. In path I, the coordinated A group on Ag$_4$P$_4$A$_2$ first reacted with DBU, and this step was slightly endergonic by 7.7 kcal/mol (Fig. 3b). Thereafter, carboxylation with CO$_2$ occurred on Ag$_4$P$_4$A$_2$-2 to generate the intermediate Ag$_4$P$_4$A$_2$c-1 (c represents CO$_2$), with a low activation barrier of 12.1 kcal/mol owing to the high nucleophilicity of the deprotonated amino group. Subsequent cyclization then occurred with a barrier of 16.3 kcal/mol. The resulting intermediate Ag$_4$P$_4$A$_2$c-2 then underwent protonation and ligand exchange to complete the catalytic cycle.

Overall, the Ag$_4$-catalyzed cycloaddition of N-benzylprop-2-yn-1-amine was highly exergonic by -37.7 kcal/mol, and the carboxylation step was the rate-determining step (Ag$_4$P$_4$A$_2$-2 → Ag$_4$P$_4$A$_2$c-1). Path II started with the coordination of an extra AH substrate, preferentially via an amino group, to form Ag$_4$P$_4$A$_3$H-1 (Supplementary Figs. 15, 16 and Fig. 3b). Similar to the overall transformation in Path I, deprotonation, carboxylation, cyclization, and protonation then occurred to generate the final product. However, the overall energy demands for Path II were 4.4 kcal/mol higher than those for Path I (26.0 vs. 21.6 kcal/mol in Fig. 3b and Supplementary Fig. 15). Of note, in this study, some other pathways, including deprotonation and carboxylation on Ag$_4$P$_4$A$_3$H-1, were also examined but were excluded because of their relatively high energy demands (Supplementary Fig. 17). In this context, Path I was the most feasible pathway. Moreover, the carboxylation process of path I was experimentally investigated by $^{13}$C NMR and ESI-MS. As shown in Supplementary Fig. 19, the $^{13}$C NMR carbon spectrum shows that the characteristic peaks of raw material **1a** gradually weakened with the insertion of carbon dioxide. Meanwhile, new peaks assigned to the products gradually emerge and enhance. The characteristic peak signal changed significantly within 0.5 h, so we monitor the ESI-MS spectrum of the reaction solution during this period. To be noted, intermediate species IV (Fig. 3) was successfully detected by ESI-MS when Ag$_4$ NC, **1a** and CO$_2$ were mixed for 15 min. The mass peak of [Ag$_4$C≡CCH$_2$NHBnC=CCH$_2$CH$_2$O$_2$NBn (Dppf)$_2$]$^+$ was detected at 1871.6 m/z (simulation: 1871.6 m/z) (Supplementary Fig. 20), coincident with the species IV on path I of DFT calculations (Fig. 3b, via

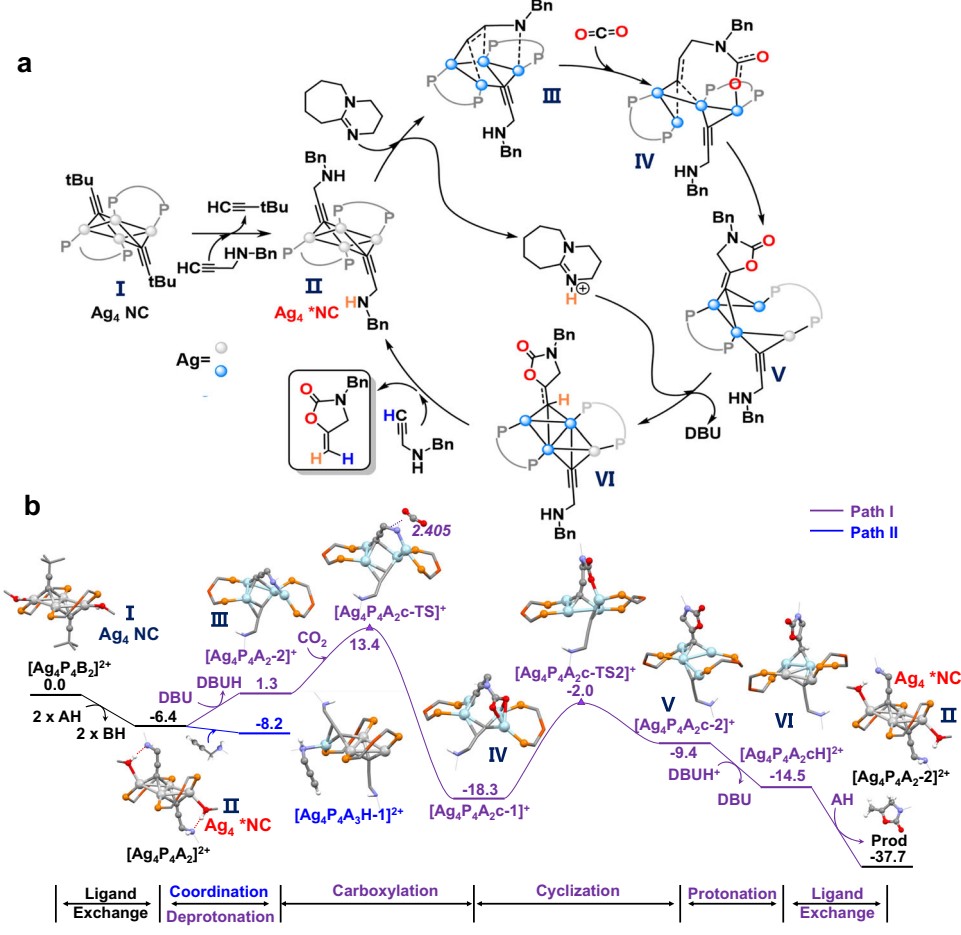

**Fig. 3 | Proposed mechanism and calculation of the relative Gibbs free energies for Ag$_4$ NC. a** Proposed mechanism by Ag$_4$ NC. **b** The relative Gibbs free energies, in bold. Gibbs free energy profiles of the Ag$_4$ NC on carboxylation of N-benzylprop-2-yn-1-amine. Abbreviated labels: AH (N-benzylprop-2-yn-1-amine,**1a**); BH (3,3-Dimethyl-1-butyne); c(CO$_2$); P$_2$(dppf). For clarity, the two MeOH molecules, all H atoms (unless the reaction site), and the benzyl group on N-benzylprop-2-yn-1-amine were omitted in all structures except for Ag$_4$P$_4$B$_2$ and Ag$_4$P$_4$A$_2$. Silver: silver and light blue.

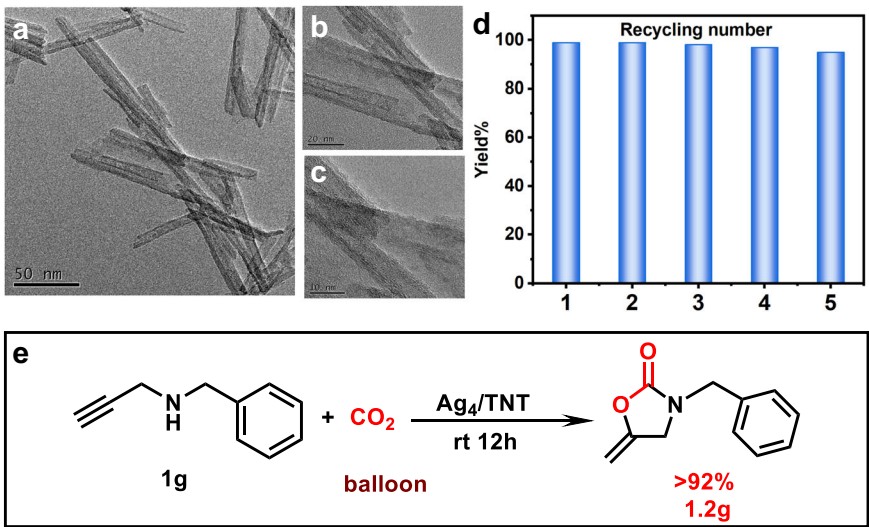

**Fig. 4 | Characterization and applications of Ag₄/TNT. a–c** TEM images of Ag₄/TNT. **d** Recoverability of Ag₄/TNT catalysts in the cyclization of propargylamine with $CO_2$. reaction conditions: Ag₄/TNT (50 mg, 1.6 wt% loading of NCs), N-benzylprop-2-yn-1-amine (0.5 mmol), DBU (0.05 mmol), acetonitrile (1.0 mL), 25 °C, 12 h, $CO_2$ balloon. **e** Gram scale experiment.

ligand exchange). The tetranuclear Ag₄ core was pivotal in stabilizing the deprotonated amino group in Ag₄P₄A₂-2 and the anionic carboxylic group in Ag₄P₄A₂c-1. Such an interaction was unlikely in the Ag₂ system, as Ag-N coordination resulted in remarkable structural distortion in the diphosphine ligand. This was also the reason the yield of the Ag₂ system was significantly lower than that of the Ag₄ system (Supplementary Fig. 18). Based on the above, we proposed a mechanism for the cyclization of propargylamine with $CO_2$ catalyzed by Ag₄ NC (Fig. 3a). Obviously, Ag₄ NC first interacted with the propylamine substrate to produce the dehydrogenation activation product Ag₄* NC, which remained in the form of Ag₄* NC after cyclization. Throughout the catalytic process, activation of the substrate required coordination between multiple Ag atoms (the blue atoms represent the active Ag atoms), confirming the reaction pathways that are inaccessible to the Ag single-atom catalyst and Ag₂ NC.

To understand its applicability, the Ag₄/TNT nanocomposite was successfully synthesized, which characterized by solid-state UV, XRD, TEM, XPS and element mapping (see Fig. 4 and Supplementary Figs. 7–11 for details). The Ag₄/TNT nanocomposites demonstrated the same activity as Ag₄ NC, while TNT carrier was inactive (Supplementary Fig. 12). In this scenario, a recycling experiment was performed with **1a** as the substrate, and the reaction efficiency did not show significant changes even after five runs (Fig. 4d and Supplementary Fig. 13). To determine the practicability of this transformation, a scale-up experiment afforded 3-benzyl-5-methylene-2-oxazolone in 1.2 g and >92% yield, which is comparable to previous results (Fig. 4e).

In summary, alkyne-protected low-nuclear Ag₄ nanocluster (NC) is designed to catalyze the cyclization of propargylamine with $CO_2$. As expected, the low-nuclear Ag₄ NC achieves the highest TON value of 5746.2, significantly higher than that of reported catalysts and the corresponding Ag₂ NC, Ag₆ NC and Ag₉ NC. In addition, the Ag₄ NC successfully achieves the cyclization of propargylamine with $CO_2$ under mild conditions. In the elementary reaction of Ag₄ NC with substrates, including HC≡CCH₂NHBn, HC≡CCH₂NHCy and HC≡CCH₂NHⁿBu, we capture three Ag₄* NC intermediates and confirm their structures by Electrospray ionization (ESI). Density functional theory (DFT) calculations further confirm the key role of four Ag atoms, revealing the feasibility and importance of low-nuclear catalysts at the atomic level. Importantly, the Ag₄/TNT (functional titanate nanotubes) nanocomposite afford the product at the gram scale.

Therefore, the customized Ag₄ catalyst improves the reaction activity while exerting the atomic economy similar to that of single atom catalyst, which has advantages in reducing cost. The present work provides a new perspective on the mechanism of the cyclization of propargylamine with $CO_2$, which provides further support for the design of further atomic level catalysts and their efficient utilization.

## Methods
### Characterizations
The UV−vis. spectra were recorded on a Techcomp UV 1000 spectrophotometer. Transmission electron microscopy (TEM) was conducted on a JEM-2100 microscope with an accelerating voltage of 200 kV. The FT-IR spectra were recorded with a Bruker Tensor 27 instrument. The X-ray diffraction (XRD) patterns were obtained on Smart Lab 9 KW with Cu Kα radiation. The NCs loaded on the TNT catalyst support were determined by Inductively Coupled Plasma Mass Spectrometry (ICP-MS). The X-ray photoelectron spectroscopy (XPS) measurements were conducted on ESCALAB 250Xi. Electrospray ionization mass spectra (ESI-MS) were recorded using a Waters UPLC H-class/Xevo G2-XS Qtof mass spectromete.

### Catalytic activity
A typical "the cyclization of propargylamine with $CO_2$" reaction was used to evaluate the catalytic performance of Ag₄ NC. Ag₄ NC (0.4 mg, $0.2 \times 10^{-3}$ mmol), propargylamines (0.5 mmol), and 1,8-Diazabicyclo [5.4.0] undec-7-ene(DBU) (0.05 mmol) were added to acetonitrile (1 mL) in the reaction tube. The reaction stirring for 2 h at 25 °C with the balloon in Carbon dioxide atmosphere. After the reaction stopped, The reaction solution was diluted by dichloromethane, The conversion and selectivity were determined by GC analysis and column chromatography (EtOAc/PE = 1:5).

## Data availability
Data supporting the findings of this work are available within the article and its Supplementary Information. The data that support the findings of this study are available from the corresponding author upon request. The X-ray crystallographic structures reported in this work have been deposited at the Cambridge Crystallographic Data Center (CCDC) under deposition numbers 2254886 for [Ag₂dppf₃]. These data can be obtained free of charge from the CCDC via https://www.ccdc.cam.ac.uk/structures/.

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

## Acknowledgements

We acknowledge financial support by the National Natural Science Foundation of China under grant number 21972001 (H.T.S.) and 21871001 (M.Z.Z.) and Natural Science Foundation of Anhui Province, Anhui University under grant number 2008085MB37(H.T.S.).

## Author contributions

L.L. performed experiments and paper writing. Y.L. performed DFT theoretical studies and paper writing. Y.L.D., H.F.L., Y.P.Y., Z.Y.Z. participated in the data analysis and revision; H.T.S., H.Z.Y., and M.Z.Z. contributed to the design of the study, data analysis, and paper writing.

## Competing interests

The authors declare no competing interests.
