## [Peer Review File · Nature Communications]

REVIEWER COMMENTS

Reviewer #1 (Remarks to the Author):

This manuscript reports a low-nuclear Ag₄ nanocluster that can effectively catalyze the cyclization of propargylamine with CO₂ under mild conditions. The corresponding TON and TOF can reach 5746.2 and 2873.1 h⁻¹, respectively, which are far higher than the reported results. The mechanism investigation revealed that [Ag₄] unit played the key role in the catalytic reaction. This work shows innovation on the regulation of catalyst. Thus, I advise this work can be published in Nat. Commun. after the following revision.

- 1) To confirm its purity, PXRD of crystal Ag₄ NC should be given in the revised version.
- 2) How about the loading amount of Ag₄ NC in TNT on catalytic performance? Authors should add the corresponding control experiments.
- 3) In Figure S8, no characteristic peaks of Ag₄ cluster can be observed. How can you confirm the formation of Ag₄/TNT?
- 4) How about the stability of Ag₄ cluster in the recycle test? ICP, Mapping, and ESI-MS can be performed after recycling.
- 5) The calculating processes of TOF and TON should be given.
- 6) In Figure 1B, the values of TON and TOF were wrongly labelled. Please modified them.
- 7) Some catalysts were applied in the cyclization of propargylamine with CO₂. The cost comparison of those catalysts and Ag₄ NC can be added to state that Ag₄ NC belongs to a customized catalyst.

Reviewer #2 (Remarks to the Author):

Li et al. reports an excellent catalyst, ligand-protected Ag₄ cluster, for cyclization of propargylamine with CO₂ under mild conditions, which makes a significant progress toward chemical utilization of the greenhouse gas CO₂. Various characterizations have been conducted in experiments to help elucidate the underlying reaction mechanism. I recommend to publish the work, but an important issue should be clarified before publication. What is the unique structural characteristics in terms of electronic or geometric property enabling the Ag₄ core to exhibit better catalytic performance than Ag₆ and Ag₉, the cluster cores of which also have potential opportunities to trigger synergistic effects between adjacent metals?

Reviewer #3 (Remarks to the Author):

The work reported the catalysis of a Ag₄ cluster co-protected by alkyne and diphosphine ligands for the cyclization of propargylamine with CO₂. The catalytic performance of the cluster was compared with Ag₂, Ag₆ and Ag₉ clusters that are protected by different ligands. Some of the reaction intermediates were captured and characterized by ESI mass and ¹H NMR. Overall, the work should be interesting for publication in Nature Communications. However, before publication, the authors should improve the manuscript by addressing the following issues:

1. The clusters studied in this work were prepared with reported methods. To enhance the novelty, the authors should synthesize more Ag₄ clusters with different ligand protections to illustrate how the surrounding ligands influence the catalysis.
2. The catalytic comparison of Ag₄ with Ag₉ is unfair because Ag₉ was fully coordinated by thiolate ligands. Therefore, more investigations should be performed to understand why Ag₂ and Ag₆ clusters also exhibited reasonable activities. In the case of Ag₆, is it possible that it is hard for the reactants to exchange the acetate ligands?
3. Ag₄ NC# should be defined in the main text. Why Ag₄ NC# exhibited much better performance than Ag₄ NC? During the catalysis the cyclization of propargylamine with CO₂, Ag₄ NC should be ligand-exchanged by the reactant. The activity difference between Ag₄ NC and Ag₄ NC# should be minor.
4. Experimental evidences should be provided for the carboxylation process.
5. The cyclization of propargylamine with CO₂ involves the binding of both alkynide and amine on Ag₄. How about the reaction between alkynes without amine moiety and CO₂?

Responses to the comments of reviewer 1:

This manuscript reports a low-nuclear Ag₄ nanocluster that can effectively catalyze the cyclization of propargylamine with CO₂ under mild conditions. The corresponding TON and TOF can reach 5746.2 and 2873.1 h⁻¹, respectively, which are far higher than the reported results. The mechanism investigation revealed that [Ag₄] unit played the key role in the catalytic reaction. This work shows innovation on the regulation of catalyst. Thus, I advise this work can be published in Nat. Commun. after the following revision.

1) To confirm its purity, PXRD of crystal Ag₄ NC should be given in the revised version.

Answer: Thanks for your advice. The purity of Ag₄ NC was analyzed by X-ray powder diffraction. By comparing the experimental data with the powder diffraction pattern simulated by the single crystal structure (Figure S7), the position and intensity of the peaks are basically the same indicating the high purity of Ag₄ NC here. This figure has been added to the supplementary information file as FigureS7.

Figure S7. XRD experimental data and simulation data of the Ag₄ NC.

2) How about the loading amount of Ag₄ NC in TNT on catalytic performance? Authors should add the corresponding control experiments.

Answer: Thanks for your advice. A series of Ag₄/TNT catalysts with different loading amounts (theoretical loading amounts of X=0.5, 1, 2, 3, 5 wt%, the catalysts are denoted as Ag₄ NC/TNT-X %) were prepared by the general impregnation method. The results showed that Ag₄/TNT-2% had the highest activity efficiency, reaching 99% yield at 2 h. (Table R1). After increasing the loading, the catalytic efficiency of Ag₄/TNT-3 % was slightly higher than that of Ag₄/TNT-2 % at

1.5 h, while the decrease in the catalytic activity of Ag₄/TNT-5 % was probably due to the aggregation of Ag₄ NC.

Table R1. Catalytic performance of Ag₄/TNT with different loadings. ^[a]

Entry	Catalyst	t [h]	Yield ^[b] [%]
1	Ag ₄ /TNT-0.5%	2	23.80
2	Ag ₄ /TNT-0.5%	8	85.90
3	Ag ₄ /TNT-1%	2	47.30
4	Ag ₄ /TNT-1%	4	90.20
5	Ag ₄ /TNT-2%	2	99.00
6	Ag ₄ /TNT-2%	1.5	71.00
7	Ag ₄ /TNT-3%	1.5	75.36
8	Ag ₄ /TNT-5%	1.5	63.00

[a] Reaction conditions: catalyst (Ag₄/TNT with different loadings), propargylamine (0.5 mmol), DBU (0.05 mmol), solvent (1 mL), 25°C, and CO₂ balloon. [b] Yields and selectivity were determined by gas chromatography and column chromatography (EtOAc/PE = 1:5).

3) In Figure S8, no characteristic peaks of Ag₄ cluster can be observed. How can you confirm the formation of Ag₄/TNT?

Answer: Thanks for your advice. The characteristic peaks of Ag₄ NC are not observed because the Ag₄ NC are very small. The diffraction peaks of corresponding particles appear only when the clusters undergo composition change and agglomeration¹⁻³.

To verify the successful synthesis of Ag₄/TNT, we used XPS to monitor the signal peak of Ag in Ag₄/TNT (Figure S10) and confirm the distribution of Ag, P and Fe elements of Ag₄ NC in the Mapping (Figure S11). At the same time, we found the coincidence of solid UV of Ag₄/TNT with liquid UV of Ag₄ NC (Figure S8) further confirming the formation of Ag₄/TNT.

Figure S8. Solid UV Absorption Spectroscopy of Ag_4/TNT 、 TNT and Liquid UV Absorption Spectroscopy of Ag_4 NC.

Figure S10. XPS spectrum of Ag_4/TNT and Ag_4 NC and TNT.

Figure S11. TEM image and element maps for an Ag_4/TNT sample: Ag, Ti, Fe, P.

4) How about the stability of Ag_4 cluster in the recycle test? ICP, Mapping, and ESI-MS can be performed after recycling.

Answer: Thanks for your advice. We confirm that the stability of Ag_4/TNT catalyst is good, mainly for the following reasons: First, the cycled Ag_4/TNT catalyst can still complete the

conversion of benzyl-propargylamine within 2 h, and the catalytic effect is almost consistent with that of the fresh catalyst; Secondly, we performed an ICP-MS test on the recovered Ag_4/TNT catalyst and found Ag content of 1.55wt % (the initial catalyst content was 1.60%); Then, we found that the solid UV signal of recovered Ag_4/TNT was consistent with liquid UV and fresh Ag_4/TNT ^{4,5}. (Figure S13) Finally, a mapping test was conducted on the recycled Ag_4/TNT catalyst, and the Ag_4 NC were still uniformly dispersed on the TNT after the reaction. (Figure S14) The above results show that the Ag_4/TNT catalyst is stable in the recycle test. However, the ESI-MS characterization cannot be performed because we cannot strip Ag_4 NC from Ag_4/TNT .

Figure S13. Solid UV Absorption Spectroscopy of Ag_4/TNT and Recycled Ag_4/TNT and Liquid UV Absorption Spectroscopy of Ag_4 NC.

Figure S14. TEM images and elemental maps of recycling Ag_4/TNT samples: Ag, Ti, Fe, P

5) The calculating processes of TOF and TON should be given.

Answer: Thanks for your advice. Examples of TOF and TON are as follows: Propylamine (1.5 mmol), 1,8-diazabicyclo [5.4.0] undec-7-ene (DBU, 0.05 mmol), and acetonitrile (1 ml) were mixed with Ag_4 NC (0.013 mol%) and stirred at room temperature in a carbon dioxide atmosphere

for 2h. After workup, 2a (1.12 mmol, yield 74.7%) was obtained. The TON was calculated by the equation (TON = Mole of products/mol of catalytic sites) and the value was given to be 5746.2. The TOF was calculated by the equation (TOF = Mole of products/mol of catalytic sites/time) and the value was given to be 2873.1h⁻¹.

6) In Figure 1B, the values of TON and TOF were wrongly labelled. Please modified them.

Answer: Thanks for your advice. We have carefully checked and revised the labeling errors in the manuscript. We changed the "TOF and TON values up to 5746.2 and 2873.1, respectively to "TON and TOF values up to 5746.2 and 2873.1h⁻¹, respectively" on page 2 and changed the "orange label" to "TON" and "blue label" to "TON" in Figure 1 B.

Figure 1. (A) Total structure of the Ag_n (n=2,4,6,9) NCs (B) TON and TOF value of different catalysts for CO₂ cycloaddition of N-benzylprop-2-yn-1-amine. Reaction conditions: Ag₄ NC (0.04 mol%), propargylamine (0.5 mmol), DBU (0.05 mmol), solvent (1 mL), 25°C and CO₂ balloon. Yields and selectivity were determined by gas chromatography. [a] propargylamine (1.5 mmol), DBU (0.15 mmol), solvent (1 mL), 25°C and CO₂ balloon. (C) The cyclization of various propargylamine with CO₂.

7) Some catalysts were applied in the cyclization of propargylamine with CO₂. The cost comparison of those catalysts and Ag₄ NC can be added to state that Ag₄ NC belongs to a customized catalyst.

Answer: Thanks for your advice. The low nuclear Ag₄ NC has excellent atomic economy and low cost, compared with expensive catalysts such as Au and Ru. For example, Ken-ichi Fujita et al.

synthesized AuNP@PAMAM/C catalyzed cyclization of propylamine with CO₂⁶, which required 1 mol% of Au loading; Fuyuhiko Inagaki et al. used [Au(dpb^F)]SbF₆ which required 0.5 mol% of loading for 48h to obtain 90% yield⁷; Seyed Mahdi Saadati utilized KCC/Salen/Ru(II) requiring 100°C reaction temperature⁸; Sk. Manirul Islam synthesized Ag@Tppa-1 and Ag@TpTa catalysts requiring a loading rate of 1.88 mol% Ag for 16h for the conversion of the substrate alkyne propylamine⁹. In this work, the Ag₄ NC catalyst requires only 0.04 mol% to achieve 99% product conversion at 25°C within 2h.

So, we have added the sentence in the conclusion on page 6: “Therefore, the customized Ag₄ catalyst improves the reaction activity while exerting the atomic economy similar to that of single atom catalyst, which has advantages in reducing cost.”

Responses to the comments of reviewer 2:

Li et al. reports an excellent catalyst, ligand-protected Ag₄ cluster, for cyclization of propargylamine with CO₂ under mild conditions, which makes a significant progress toward chemical utilization of the greenhouse gas CO₂. Various characterizations have been conducted in experiments to help elucidate the underlying reaction mechanism. I recommend to publish the work, but an important issue should be clarified before publication. What is the unique structural characteristics in terms of electronic or geometric property enabling the Ag₄ core to exhibit better catalytic performance than Ag₆ and Ag₉, the cluster cores of which also have potential opportunities to trigger synergistic effects between adjacent metals?

Answer: Thank you for your advice. Based on the geometric structures, the variability of the Ag₄-parallelogram structure helps to provide space for binding of the alkynyl substrate/CO₂. During the deprotonation and carboxylation steps (Figure R1), the Ag₄ NC structure changes from planar into distorted butterfly-like and then back to square planar to accommodate the π-coordination of alkynyl and σ-coordination with amine group in [Ag₄P₄A₂-2]⁺ and [Ag₄P₄A₂c-TS]⁺ and the carboxylic group in [Ag₄P₄A₂c-1]⁺ (Figure R1 AB). By contrast, the rigidity of the octahedral Ag₆ NC (Figure R2) precludes the structural distortion (and thus the extra-stability), and the strong Ag-SR bonding on Ag₉ NC prohibits the binding of the alkynyl substrate/CO₂.^{10,11}.

Figure R1. (A) Scheme of deprotonation of $[Ag_4P_4A_2]^{2+} \rightarrow [Ag_4P_4A_2]^+$ (B) Scheme of carboxylation of $[Ag_4P_4A_2]^+ \rightarrow [Ag_4P_4A_2c-1]^+$ (C) Proposed mechanism by $Ag_4 NC$.

Figure R2. The geometrical structures of $\text{Ag}_4 \text{ NC}$ and $\text{Ag}_6 \text{ NC}$

Responses to the comments of reviewer 3: The work reported the catalysis of a Ag_4 cluster co-protected by alkyne and diphosphine ligands for the cyclization of propargylamine with CO_2 . The catalytic performance of the cluster was compared with Ag_2 , Ag_6 and Ag_9 clusters that are protected by different ligands. Some of the reaction intermediates were captured and characterized by ESI mass and ^1H NMR. Overall, the work should be interesting for publication in Nature Communications. However, before publication, the authors should improve the manuscript by addressing the following issues:

1. The clusters studied in this work were prepared with reported methods. To enhance the novelty, the authors should synthesize more Ag_4 clusters with different ligand protections to illustrate how the surrounding ligands influence the catalysis.

Answer: Thank you for your advice. We tried other different approaches, such as replacing tert-butynyl silver in $\text{Ag}_4 \text{ NC}$ with cyclohexyl thio silver and isopropyl thio silver, but only synthesized some mixtures and $\text{Ag}_2 \text{ NC}$.

In order to understand the reason for the high activity of Ag₄ NC, we first selected Ag_n (n = 2, 4, 6 and 9) for comparison, and found that the activity of Ag₄ NC was higher than that of Ag₂ NC and Ag₆ NC, Ag₉ NC almost no activity. Moreover, it was found that Duan et al. synthesized TOS-Ag₄ NC in 2018, which can also catalyze the reaction of propargylamine with carbon dioxide with TON values up to 930¹², which is comparable to the Ag₆ NC and Ag₂ NC in our work and lower than that of Ag₄ NC. Based on this, we believe that the atomic number of Ag_n NC is not a decisive factor affecting the reactivity.

Ag₄ NC is the only cluster with alkyne ligands. Therefore, we speculate that ligands have an important effect on the performance. Scott et al. reported that alkyne-protected Cu₂₀ NCs do not require harsh pretreatment during catalysis¹³, Wang et al. reported that an alkyne-protected Au₃₈ NC exhibited superior performance compared to that of a sulfate-protected Au₃₈ NC¹⁴. And the activity of TOS-Ag₄ NC with Ag-S bonds is lower than that of Ag₄ NC. Unlike the case of sulfates, alkyne ligands are more readily released from metal nanoclusters than others, while maintaining the overall structure, and thus can serve as an ideal system for understanding the effect of surface stabilizers on catalytic facilitation¹¹.

This work takes advantage of the specificity of the alkyne ligand and customized the Ag₄ NC catalyst to the cyclization of propargylamine with CO₂. On the one hand, the alkyne ligand in Ag₄ NC allowed the dehydrogenation of the substrate propargylamine and the σ -activation of the Ag-C bond to proceed simultaneously, thus activating both the catalyst and the substrate and ensuring that the catalytic reaction proceeded efficiently. In addition, the captured key intermediate Ag₄ NC* played a key role in the study of the reaction performance and catalytic mechanism. On the other hand, the variable core of Ag₄ NC planar tetragonal shape, combined with the alkyne ligand which is prone to exchange, results in high structural variability. Based on this, the alkyne ligand and its variable core of Ag₄ NC play the important role in catalysis.

2. The catalytic comparison of Ag₄ with Ag₉ is unfair because Ag₉ was fully coordinated by thiolate ligands. Therefore, more investigations should be performed to understand why Ag₂ and Ag₆ clusters also exhibited reasonable activities. In the case of Ag₆, is it possible that it is hard for the reactants to exchange the acetate ligands?

Answer: Thank you for your advice. As both Ag atoms in Ag₂ NC are Ag(I), and are away from each other, we anticipate that Ag₂ NC may undergo a catalytic cycle similar to Ag salts¹⁵, through substrate coordination, deprotonation, carboxylation and catalyst regeneration steps. Meanwhile, due to the metallic core structure, the reaction of Ag₆ NC might be similar to that of Ag₄ NC to undergo binding of the alkynyl substrate, deprotonation, carboxylation, and catalyst regeneration steps. Nevertheless, the primary binding step of Ag₆ NC is significantly more difficult than that of Ag₄ NC, as the coordination of either triple bond or the amine group to the Ag site (after one P-dissociation, Figure R3a) is endergonic by 15.5 and 14.3 kcal/mol. Of note, the replacement of the carboxylic group¹⁶ on the Ag₆ NC catalyst is significantly more energy demanding (Figure R3b). By contrast, the incorporation of alkynyl substrate on Ag₄ NC is slightly exergonic by -6.4 kcal/mol.

Figure R3. the substrate exchange of acetate on Ag₆ NC.

3. Ag₄ NC# should be defined in the main text. Why Ag₄ NC# exhibited much better performance than Ag₄ NC? During the catalysis the cyclization of propargylamine with CO₂, Ag₄ NC should be

ligand-exchanged by the reactant. The activity difference between $\text{Ag}_4 \text{ NC}$ and $\text{Ag}_4 \text{ NC}^\#$ should be minor.

Answer: Thank you for your advice. In Figure 1, we changed " $\text{Ag}_4 \text{ NC}^\#$ " to " $\text{Ag}_4 \text{ NC}^{[a]}$ ". In order to evaluate the catalytic performance of $\text{Ag}_4 \text{ NC}$, we obtained TON/TOF values in here with scaled-up substrate amounts. However, $\text{Ag}_4^* \text{ NC}$ is an intermediate produced during the cyclization of propargylamine with carbon dioxide, and as you pointed out, we experimentally demonstrated that the difference in activity between $\text{Ag}_4 \text{ NC}$ and $\text{Ag}_4 \text{ NC}^*$ is small. On Page 3 of the manuscript, it is mentioned that "we isolated and verified the activity of $\text{Ag}_4^* \text{ NC}$ (R_1 =benzyl). The experimental results showed that the activities of $\text{Ag}_4^* \text{ NC}$ and $\text{Ag}_4 \text{ NC}$ are similar, confirming that $\text{Ag}_4^* \text{ NC}$ is the key intermediate".

Figure 1. (A) Total structure of the Ag_n (n=2,4,6,9) NCs (B) TON and TOF value of different catalysts for CO_2 cycloaddition of N-benzylprop-2-yn-1-amine. Reaction conditions: $\text{Ag}_4 \text{ NC}$ (0.04 mol%), propargylamine (0.5 mmol), DBU (0.05 mmol), solvent (1 mL), 25°C and CO_2 balloon. Yields and selectivity were determined by gas chromatography. [a] propargylamine (1.5 mmol), DBU (0.15 mmol), solvent (1 mL), 25°C and CO_2 balloon. (C) The cyclization of various propargylamine with CO_2 .

4. Experimental evidences should be provided for the carboxylation process.

Answer: Thank you for your advice. The calculations demonstrate that the carboxylation occurs on the incoming A-substrate (path I) rather than extra AH substrate (Path II).

According to the suggesting, we designed relevant experiments to verify it. Firstly, the carboxylation process of path I was experimentally investigated by ^{13}C NMR and ESI-MS. As shown in Figure S19, the ^{13}C NMR carbon spectrum shows that the characteristic peaks of raw material 1a gradually weakened with the insertion of carbon dioxide. Meanwhile, new peaks assigned to the products gradually emerge and enhance. The characteristic peak signal changed significantly within 0.5 h, so we monitor the ESI-MS spectrum of the reaction solution during this period. To be noted, intermediate species IV (Figure 3) was successfully detected by ESI-MS when Ag_4NC , 1a and CO_2 were mixed for 15min. The mass peak of $[\text{Ag}_4\text{C}\equiv\text{CCH}_2\text{NHBnC}=\text{CCH}_2\text{CH}_2\text{O}_2\text{NBn}(\text{Dppf})_2]^+$ was detected at 1871.6 m/z (simulation: 1871.6 m/z) (Figure S20), coincident with the species IV on path I of DFT calculations (Figure 3B, via ligand exchange). The above result was added to page 5 of the revised manuscript.

Figure S19 ^{13}C NMR spectra monitoring of CO_2 cycloaddition of N-benzylprop-2-yn-1-amine by Ag_4NC .

Figure S20 (A) ESI-MS spectra of intermediates in the reaction process and simulation of the corresponding mass spectra. (B) the carboxylation process of path I and path II.

5. The cyclization of propargylamine with CO₂ involves the binding of both alkynide and amine on Ag₄. How about the reaction between alkynes without amine moiety and CO₂?

Answer: Thank you for your advice. We have tried Ag₄/TNT to catalyze the carboxylation of phenylacetylene with CO₂ (Figure R4 (I)) and the cyclization of 2-Methyl-3-butyn-2-ol with CO₂ (Figure R4(II)), both of which are amine-free alkynes. The results showed that Ag₄/TNT had good performances under different reaction conditions (Table R2).

Table R2. Catalytic performance of Ag₄/TNT in the cyclization reaction of alkynes without amine groups and CO₂.

Reaction	Catalyst	Solvent	Base	t [h]	Yield[%]
I [a]	Ag ₄ /TNT	MeCN	DBU	6	99.00
II [b]	Ag ₄ /TNT	DMSO	Cs ₂ CO ₃	12	80.60

Reaction conditions: [a]Ag₄/TNT (50 mg) (1.60 wt% loading of Ag₄ NC, ICP), 2-Methyl-3-butyn-2-ol (1 mmol), DBU (1 mmol), MeCN (1 mL), CO₂ balloon, Rt, 6h. Yields and selectivity were determined by gas chromatography. [b]Ag₄/TNT (50 mg) (1.60 wt% loading of Ag₄ NC, ICP), alkyne (1 mmol), Cs₂CO₃ (0.24 mmol), DMSO (1 mL), CO₂ (1.0 bar), 40 °C, 12h. Yield of isolated product.

Figure R4. Ag₄/TNT catalyzed reaction of different alkynes with CO₂.

Reference:

- Ding, T. *et al.* Atomically Precise Dinuclear Site Active toward Electrocatalytic CO₂ Reduction. *Journal of the American Chemical Society* **143**, 11317-11324.
- Fang, Y. *et al.* Insight into the Mechanism of the CuAAC Reaction by Capturing the Crucial Au₄Cu₄- π -Alkyne Intermediate. *Journal of the American Chemical Society* **143**, 1768-1772 (2021).
- Yun, Y. *et al.* Exploiting the Fracture in Metal-Organic Frameworks: A General Strategy for Bifunctional Atom-Precise Nanocluster/ZIF-8(300 °C) Composites. *Small* **18**, 2107459 (2022).
- Sullivan, A. I. *et al.* Synthesis and Characterization of a Monodentate N-Heterocyclic Carbene-Protected Au₁₁-Nanocluster via Reduction with KC₈. *Chemistry of Materials* **35**, 2790-2796 (2023).
- Lummis, P. A. *et al.* NHC-Stabilized Au₁₀ Nanoclusters and Their Conversion to Au₂₅

- Nanoclusters. *JACS Au* **2**, 875-885 (2022).
6. Matsuo, H., Fujii, A., Choi, J.-C., Fujitani, T. & Fujita, K.-i. Carboxylative Cyclization of Propargylic Amines with Carbon Dioxide- Catalyzed by Polyamidoamine-Dendrimer-Encapsulated Gold Nanoparticles. *Synlett* **30**, 1914-1918 (2019).
 7. Inagaki, F., Maeda, K., Nakazawa, K. & Mukai, C. Construction of the Oxazolidinone Framework from Propargylamine and CO₂ in Air at Ambient Temperature: Catalytic Effect of a Gold Complex Featuring an L₂/Z-Type Ligand. *European Journal of Organic Chemistry* **2018**, 2972-2976 (2018).
 8. Saadati, S. M. & Sadeghzadeh, S. M. KCC-1 Supported Ruthenium-Salen-Bridged Ionic Networks as a Reusable Catalyst for the Cycloaddition of Propargylic Amines and CO₂. *Catalysis Letters* **148**, 1692-1702 (2018).
 9. Ghosh, S. *et al.* Utility of Silver Nanoparticles Embedded Covalent Organic Frameworks as Recyclable Catalysts for the Sustainable Synthesis of Cyclic Carbamates and 2-Oxazolidinones via Atmospheric Cyclizative CO₂ Capture. *ACS Sustainable Chemistry & Engineering* **8**, 5495-5513 (2020).
 10. Wang, Y. *et al.* Atomically Precise Alkynyl-Protected Metal Nanoclusters as a Model Catalyst: Observation of Promoting Effect of Surface Ligands on Catalysis by Metal Nanoparticles. *Journal of the American Chemical Society* **138**, 3278-3281 (2016).
 11. Yan, J., Teo, B. K. & Zheng, N. Surface Chemistry of Atomically Precise Coinage-Metal Nanoclusters: From Structural Control to Surface Reactivity and Catalysis. *Accounts of Chemical Research* **51**, 3084-3093 (2018).
 12. Chang, Z., Jing, X., He, C., Liu, X. & Duan, C. Silver Clusters as Robust Nodes and π -Activation Sites for the Construction of Heterogeneous Catalysts for the Cycloaddition of Propargylamines. *ACS Catalysis* **8**, 1384-1391 (2018).
 13. Cook, A. W., Jones, Z. R., Wu, G., Scott, S. L. & Hayton, T. W. An Organometallic Cu₂₀ Nanocluster: Synthesis, Characterization, Immobilization on Silica, and "Click" Chemistry. *Journal of the American Chemical Society* **140**, 394-400, (2018).
 14. Wan, X.-K., Wang, J.-Q., Nan, Z.-A. & Wang, Q.-M. Ligand effects in catalysis by atomically precise gold nanoclusters. *Science Advances* **3**, e1701823.
 15. Yoshida, M., Mizuguchi, T. & Shishido, K. Synthesis of Oxazolidinones by Efficient Fixation of Atmospheric CO₂ with Propargylic Amines by using a Silver/1,8-Diazabicyclo[5.4.0]undec-7-ene (DBU) Dual-Catalyst System. *Chemistry – A European Journal* **18**, 15578-15581 (2012).

REVIEWERS' COMMENTS

Reviewer #1 (Remarks to the Author):

I have read the revised version and response letter carefully. Authors well explained/answered all issues I was concerned in the revised manuscript,so I recommend to accept this work as it is.